

# Brief communication: A technique for making in-situ measurements at the ice-water boundary of small pieces of floating glacier ice

Hayden Johnson[1], Oskar Glowacki[2], Grant Deane[1], and Dale Stokes[1]

[1]Scripps Institution of Oceanography, University of California San Diego, 9500 Gilman Drive, La Jolla, CA 92093, United States
[2]Institute of Geophysics, Polish Academy of Sciences, 64 Księcia Janusza Str. 01-452 Warsaw, Poland

**Correspondence:** Dale Stokes (dstokes@ucsd.edu)

**Abstract.**

This paper presents a method for making direct in-situ measurements of the ice-water boundary of small pieces of floating glacier ice. The method involves approaching ice pieces in a small boat and attaching instruments to them using ice screws. These types of measurements provide an opportunity to study small-scale processes at the ice-water interface which control heat flux across the boundary. Recent studies have suggested that current parameterizations of these processes may be performing poorly. Improving understanding of these processes may allow for more accurate theoretical and model descriptions of submarine melting.

## 1 Introduction

This paper presents a technique for making in-situ measurements at the near ice-ocean boundary layer of small pieces of floating glacier ice (termed growlers). Here, the near ice-ocean boundary, or proximal boundary, refers to the region within 1 m of the ice surface. On these short scales, the behavior of the proximal boundary of growlers may be representative of other glacier ice-ocean boundaries subjected to similar forcing, such as the near-surface terminus of a glacier not subjected to plume forcing. The attention of this paper is restricted to vertical ice surfaces and is motivated by the role of the proximal boundary in melting glacier ice.

The proximal boundary consists of a solid-liquid interface, complicated by the presence of salts in the water and pressurized gas that moves from the ice into the water. The class of problems describing the diffusion of heat in a medium accompanied by a change of phase state are referred to as Stefan problems (Rubinštein, 1971). This particular Stefan problem is complicated by the presence of salt and gas, and processes that play out across a wide range of scales, as shown in figure 1. Theoretical frameworks to describe this interface are an active area of research. Current theoretical descriptions are based on conservation of heat and salt at the ice-water interface (Hellmer and Olbers, 1989; Holland and Jenkins, 1999). This description is combined with buoyant plume theory to describe the effects of rising meltwater and, in the case of glacier termini, subglacial discharge water on the fluxes across the proximal boundary (Jenkins, 2011; Cowton et al., 2015; Slater et al., 2016). This framework is largely untested by direct observations (Straneo and Cenedese, 2015; Cenedese and Straneo, 2023), and does not include gas effects. A large number of laboratory experiments have been conducted to study processes occurring at the ice-water





interface (McCutchan and Johnson, 2022). Many of these incorporate the effects of both heat and salt, but realistic effects of the pressurized bubbles present in glacier ice are difficult to reproduce in the laboratory. Wengrove et al. (2023 (accepted) have suggested that such processes could impact the boundary layer and affect heat transport through the proximal boundary. Results from numerical modelling studies on the scale of the proximal boundary have shown general agreement with some laboratory studies (Wells and Worster, 2011; Gayen et al., 2016), but these models also typically neglect the effects of bubbles.

There are indications that the description of fluxes across the ice-water boundary currently in use need to be improved in order to allow for accurate estimates of submarine melting. Recent observational studies comparing plume theory derived estimates of subglacial melting with direct observations of terminus ablation from acoustic surveys (Sutherland et al., 2019) and estimates of melting based on hydrographic data (Jackson et al., 2020) have suggested that the current theoretical framework underestimates submarine melting outside of subglacial discharge plumes. Laboratory studies have also indicated that the

existing framework may perform poorly when water velocities are slower (McConnochie and Kerr, 2017).

    Improving the description of fluxes across the ice-water boundary will require a combination of theoretical studies, laboratory experiments, and field observations. Several recent papers have suggested adjustments to the existing parameterizations in order to better match field observations (Jackson et al., 2022; Schulz et al., 2022). This is an excellent step, and should be supported by theoretical and laboratory studies. Direct observations of the proximal boundary at glacier termini are likely to remain

elusive, due to the difficult and dangerous nature of conducting measurements near calving fronts of glaciers (Herrero, 2018). Recently, robotic vehicles have been used to make observations of water properties close to the terminus (Slater et al., 2018; Poulsen et al., 2022), but making prolonged measurements in the proximal boundary layer at glacial termini or large icebergs remains a challenge. Observations of the proximal boundary at small growlers can provide a natural laboratory, and a bridge to better understanding of the physical processes that may occur there. This paper describes a method by which such observations

can be made, which we have termed "the ice frame", and briefly presents some example data that was collected using this method.

## 2   Methods

### 2.1   Description of the ice frame

The ice frame consisted of a collection of aluminum tubes which could be fastened together using adjustable clamps. Ice screws

were used to fasten the frame to a growler, and instruments could be attached to the frame using clamps and held within the proximal boundary. Figure 2a shows a schematic of the most successful configuration of the frame, and figure 2b is a picture of the frame deployed on a floating growler. Details of the specific hardware elements of the ice frame are given in appendix A. A camera, hydrophone, and micro thermistor array were deployed on the frame. A detailed description of those sensors is given in appendix B.



## 2.2 Design principles

There were four key principles that guided the design of the ice frame and the instruments deployed on it. First and foremost, the frame had to be simple to deploy. Attaching equipment to a piece of floating ice from a small boat is a difficult feat even in ideal conditions. Using only two ice screws, and placing them above the waterline, was key to keeping the frame deployment simple. Earlier iterations of the frame that included three or four points of contact with the ice, above and below the waterline, worked well during laboratory testing but proved to be too difficult to work with in the field. Second, the frame had to be adjustable in order to accommodate the natural variability in the surface of growlers. However, it also had to be easy to lock all of the moving parts into place once the frame was positioned correctly, in order to hold the instruments fixed relative to the ice face. Third, the frame, and instruments deployed on it, had to be sufficiently robust to hold up to inevitable bumping and banging during deployment and recovery. Finally, the components of the frame, and the instruments, had to be inexpensive and replaceable, because there was a non-trivial chance that the frame and any instruments on it could be damaged or lost during the deployment.

## 2.3 Deployment

The field campaign described here was conducted in Hornsund Fjord, Svalbard. The Polish Polar Station, which is located on the north side of the fjord near the mouth, was used as a base of operations for the duration of the campaign. The frame was deployed from a zodiac crewed by 2-3 people, on growlers that were floating in the water. Some of the growlers were in the bay of Hansbreen glacier, which is about 2 km northeast of the station, and others were near the northern shore of the main arm of the fjord, between the station and Hansbreen.

Taking as many steps as possible to prepare the frame for the deployment before setting out in the boat was important for ensuring that the frame could be deployed quickly and smoothly. This included readying all of the instruments, and pre-assembling the frame, in particular the connection between the cross beam and main vertical beam (see figure 2 a).

After setting out in the boat, a growler was selected on which to deploy the frame. There were several criteria involved in growler selection, the most important being location. The chosen growler had to be a safe distance (at least a few hundred meters) from the glacier terminus and from any large icebergs, so that calving events or iceberg rolling or disintegrating would not pose a danger to the boat crew. It also had to be sufficiently far from shore that the boat would not be at risk of running aground. Attention had to be payed to the surface currents and wind, to predict whether a growler that was in a safe spot at the time of selection might move to an unsafe location over the expected duration of the deployment. Aside from safety concerns, it was also important to choose a growler that was fairly isolated, and not surrounded by thick melange or other growlers that would make navigating the boat difficult and could potentially damage the frame while it was deployed.

The other selection criteria were related to the size and shape of the growler itself. Deployment and recovery of the frame required the boat to be parked up against the growler for several minutes, with crew members leaning over the side of the boat holding it in position, fastening ice screws, and adjusting various elements of the frame. This involves an element of risk, which is exacerbated by the potential for floating ice to behave unpredictably, either rolling over or breaking apart. One way





in which this risk was mitigated was by selecting growlers of appropriate size. The growler had to be large enough for the ice frame to fit onto it, and present an approximately vertical face extending at least a half a meter below the waterline and about 30 cm above it, but not so large as to pose a danger to the boat and crew in the event that it might suddenly break apart or roll over. The growlers that were selected were typically about 3-5 m in length and width, with a surface expression of about half a meter.

In addition to size, the geometry of the growlers played a role in the selection process. Growlers that were compact and looked sturdy were preferred over those that had protrusions or weak points that might have broken and caused sudden movement. Among growlers that seemed unlikely to break apart, it was also a priority to select those which seemed to be stable in the water, and not likely to roll over. This was assessed by observing the response of the growler to the ambient wave field for a few moments, and choosing growlers that were less affected. Sometimes a growler would appear to be stable based on this initial visual assessment, but it would become clear once we started attaching the frame that it was prone to rolling, and it would have to be abandoned and the selection process would start over.

Once a growler was selected, the the boat was maneuvered alongside the face where the frame was to be deployed. One crew member would hold on to the growler, if it had a useful spot to grip onto, or, if not, would fasten an extra ice screw to use as a anchoring point to hold on to, in order to keep the boat in position. The two ice screws to which the frame would be attached were then screwed into the ice. The instruments, and sometimes the additional standoff beam, were attached to the frame and adjusted, based on visual estimation, to try to accommodate the geometry of the ice face to which the frame would be attached. A line was attached to the frame at one end, and the safety floats at the other, and the floats were then placed in the water, while taking care to ensure the line would not become tangled with anything or anyone on the boat. The frame was then lowered over the side of the boat, between the boat and the growler, and the crossbeam was then attached to each of the two ice screws using pairs of SmallRig clamps. The positioning of the frame and instruments was then assessed, and if the instruments or frame were not in the right position, they were adjusted in place if possible, or else by removing the frame from the water and making adjustments on the boat before re-deploying it. This process was sometimes repeated several times before a satisfactory configuration was achieved. See figure 2 for a schematic and image of the frame.

Once the frame was successfully deployed, the boat would be moved some distance away (about 50-100 m) in order to minimize any unnecessary impacts on the environment that might influence the measurements, but would be kept close enough to monitor the growler and frame. Once enough data had been gathered, the frame would be recovered in a similar manner to that in which it was deployed. The boat would be maneuvered up to the frame, which would then be detached from the ice screws and lifted into the boat. The safety floats would then be recovered. Finally, the ice screws would be removed from the growler.





# 3 Results

## 3.1 Performance

As a mounting system, the ice frame has length, time, payload, and placement performance specifications. The frame is deployable on growlers ranging in size from 3 - 5 m across and provides access to the top 0.5 - 1 m of the ice-water interface. Instruments can be placed from the ice face up to about 0.5 m away. The time that the frame can be effectively deployed for is limited to about 1 - 2 hours because melting around the ice screws loosens the frame attachment points. Practical experience with the frame and a boat suggests that payloads up to about 5 kg can be managed. Three different types of data were success-
fully collected using the frame: images of the ice face, acoustic recordings, and temperature array measurements. These are described in greater detail below.

## 3.2 Imaging of the ice face

The frame is well-suited to collecting direct-view images of the ice at close range. Using a camera mounted on the frame and positioned about 12 cm from the ice face, images of the ice face were successfully collected during this deployment. The
camera resolution of about 25 um per pixel at the ice face (see appendix B1 for camera specifications) was sufficient to resolve individual bubbles in the ice and in the water. An example image is shown in figure 3a, in which bubbles in the ice are clearly visible. The frame rate was also high enough to capture a rising bubble in several frames as it traversed the field of view, enabling calculation of the rise velocity.

It should be noted that, since the index of refraction for ice is much closer to that of water than to that of air, it is easy to view
bubbles within the ice, but difficult to discern the ice-water boundary itself in direct-view images. A wider field of view could be obtained by mounting the camera further from the ice but this would create some difficulties: first, water clarity potentially becomes an issue, and second, the logistics of attaching a sensor to the frame increases in difficulty the further from the ice the sensor is.

## 3.3 Acoustic recordings

The ice frame allowed the collection of acoustic data at close proximity to the ice face of a melting growler. (See appendix B1 for instrumentation details.) An example of acoustic data collected from the middle of the fjord is presented in figure 3f-e. The release of bubbles from melting ice, at the terminus and floating in the bay, is responsible for most of the sound that is present in this recording. The recorded signal, which is the result of the superposition of many simultaneous releases and the propagation of this sound through the bay, is broadband and stationary, resembling white noise. Data collected from a hydrophone mounted
on the frame 5-10 cm from the face of the ice is shown in figure 3c-d. In the acoustic data collected from the frame, the impulsive nature of the sound is clearly discernible. Such data allows for the study of noise produced by ice melting in-situ at the level of individual bubble releases, thus making it easier to understand and characterize the acoustic signal of melting before it becomes distorted by propagation and superposition.



One difficulty that should be noted is that placing a hydrophone in such close proximity to the ice face results in absolute

sound pressure levels that are too high for many off-the-shelf hydrophones to measure, resulting in clipped signals. Care needs to be taken to ensure that the sensitivity of the hydrophone and recording system is not too high. The hydrophone used in this study had a sensitivity of -208 dB re 1 Vrms per $\mu$Pa, and it was used with no pre-amplifier.

## 3.4 Temperature measurements

The ice frame allowed for the successful collection of in-situ measurements of water properties in the proximal boundary.

Using an array of calibrated thermistors (see appendix B2 for details), the profile of the water temperature orthogonal to the ice face about 30 cm below the surface of a floating growler was measured over a span of about 20 minutes. A contour plot of a subset of this temperature data is shown in figure 3a. Notably, the temperature is still in excess of 1 °C at the closest sensor to the ice face, which was about 4 mm from the face. While only temperature was measured in this study, in principle other properties of the water within about 0.5 m of the ice-water boundary, such as salinity, velocity, and turbulence, could also be

measured using the ice frame.

## 4 Discussion and conclusions

The ice frame allows for direct in-situ measurements at the ice surface, and in the water up to about 0.5 m from the ice, at floating growlers. There are obvious differences between the ice-ocean boundary at floating growlers and at a glacier terminus, such as the vertical scale of the ice wall and the potential presence of subglacial discharge plumes. However, some of the

physical processes and dynamics on sub-meter scales may be common between the two environments. These may include the release of air bubbles into the ice, or other processes that modulate heat flux within the proximal boundary. Floating growlers may therefore be an effective natural laboratory that can be used to study ice-ocean interactions at these scales, with relevance to glacial termini. This boundary is more realistic than one that can be recreated in the lab with artificial ice, and also more easily accessible than the terminus itself.

The study of floating ice in glacial bays is also consequential in its own right. This ice can play an important role in controlling conditions in glacial fjords (Davison et al., 2020). Understanding small-scale processes that modulate the interaction of floating ice with water in the fjord can aid understanding of the fjord-scale influence of floating ice.

Extension of the ice frame, or similar techniques, to allow for deployment at the terminus itself, and at greater depths below the surface, is a natural future direction of research. The risk of calving icebergs means that any such deployment will

need to be carried out with unmanned robotic vehicles. Unmanned vehicles have been used recently to conduct surveys near glacial termini (Slater et al., 2018), but not to make measurements within the proximal boundary itself. Making these types of measurements is challenging, but may be necessary for the effort to accurately describe fluxes and melting at the termini of tidewater glaciers.





*Data availability.* The data presented in this paper have been uploaded to the National Science Foundation Arctic Data Center, and are
awaiting approval for publication before they are made publicly available and assigned a permanent DOI.

## Appendix A: Frame design

The ice frame consisted of several components, each of which is identified in Figure 2a. The main structure was made up of hollow, 1 inch outer diameter cylindrical aluminum tubes. The tubes were held together with two different types of clamps. Off-road light bar mounting brackets, manufactured by Mosleyz, were used for joints that did not need to be adjusted in the

field. Pairs of SmallRig Super Clamps, fastened to either end of a 3/8" threaded rod, were used for connections that needed to be made or adjusted during the deployment process. Two Petzl 19 cm ice screws were used to fasten the frame to the ice. Finally, a pair of flotation spheres were used to prevent the frame from sinking. Marine grease was used to lubricate and protect any corrosion-sensitive elements, such as nuts, bolts, and threaded rods.

The ice screws were screwed into the growler face about 20 cm above the waterline, with a horizontal separation of about 60

cm. A horizontal crossbeam was then attached to both screws with pairs of adjustable SmallRig clamps. A vertical beam with a length of 60 cm was fastened to this crossbeam at a right angle using the Mosleyz bar mounting brackets, and extended down below the surface of the water. The instruments were attached to the bottom of this vertical beam using pairs of adjustable SmallRig clamps. In some cases, the instruments themselves formed the third point of contact with the ice face. In other cases, an additional aluminum tube was connected to the bottom of the vertical beam and formed the third point of contact with the

ice face, and the instruments were either attached to this tube or to the vertical beam. The flotation spheres were tied to one end of a 5-10 m long rope, with the other end tied to the main structure of the frame.

## Appendix B: Instruments

### B1  Hydrophone and camera

An underwater camera and a hydrophone were used to make coincident optical and acoustic observations of air bubbles being

released from the ice. The camera was a GoPro Hero 5, modified using a Back-Bone lens adapter and a Schneider 17 mm lens. A resolution of 1080p and framerate of 120 frames per second were used. A Voltaic V25 USB battery pack was used to extend the battery life of the GoPro. The camera and the battery pack were contained in a 4-inch diameter watertight enclosure from Blue Robotics. The enclosure was attached to the bottom of the vertical beam of the frame using a SmallRig clamp on one end of a threaded rod, with the other end screwed into a plate which was mounted to the back end of the enclosure using Blue

Robotics enclosure clamps. A pair of wooden rods were fastened to the sides of the enclosure, extending out past the front face to the focal plane of the camera, which was about 12 cm beyond the front face of the enclosure. When the frame was deployed on a growler, the rods would rest against the submerged ice face and hold the camera an appropriate distance from the ice face such as to place it within the focal plane of the camera. A GoPro Smart Remote was used to turn the camera on and start recording before putting the frame in the water, allowing the watertight enclosure to remained sealed.





An AS-1 hydrophone, from Aquarian Scientific, was used to make acoustic recordings at close proximity to the ice face. The sensitivity of the AS-1 hydrophone was $-208$ dB re 1 V per $\mu$Pa. The hydrophone was mounted on one of the wooden standoff rods, about 7 cm from the ice face. The hydrophone cable was cut to a length of about 2 meters and spliced to an Impulse underwater connector. The recording system for the hydrophone consisted of a Tascam DR40-X handheld recorder, with a unity-gain, high-impedence buffer between the hydrophone output and the Tascam input. The recording system was housed in a 3-inch diameter Blue Robotics watertight enclosure. An external switch was connected to an Adafruit Feather microcontroller which emulated a Tascam RC-10 remote control (see github repository https://github.com/abbrev/tascam-rc-10-remote), allowing the Tascam to be turned on and start recording without opening the enclosure. The video and audio data from the camera and hydrophone were synchronized by starting recording of both instruments, then placing the hydrophone in the field of view of the camera and tapping it.

## B2 Thermistors

The second instrument configuration that was successfully deployed during this campaign was a thermistor array. The array consisted of 8 individual MF58 3950B 10 k$\Omega$ glass-encapsulated thermistors. The thermistors were glued to a 2 mm diameter carbon fiber rod. A silicone spacer about 2 mm thick was placed between the rod and each thermistor in order to help isolate the elements from the thermal mass and conductivity of the rod. AWG 30 wires were soldered to the ends of the thermistors. A common ground wire, and 8 signal wires, ran from the array to a Blue Robotics watertight enclosure containing the acquisition electronics. The exposed solder joints of the array were coated with polyurethane and then silicone in order to make the array waterproof. The data was recorded by an Adafruit Feather microcontroller, using two ADS1115 16-bit 4-channel analog-to-digital converters. For each channel, the output of a voltage regulator was connected by a 27.2 k$\Omega$ resistor to the appropriate analog-to-digital input, which was then connected to one of the resistor signal wires.

During deployment, the thermistor array was fastened to the bottom of the vertical beam of the frame using a pair of SmallRig clamps. The array was positioned to extend out from the frame towards the growler, perpendicular to the ice face. An aluminum rod was also connected to the bottom of the vertical beam, extending out towards the ice face but at an angle of about 20 degrees below horizontal. The rod and array were adjusted such that the rod would support the weight of the frame against the ice, while the array would just barely touch the ice face.

The Steinhart-Hart equation was used to compute the estimated temperature of each thermistor given the measured resistance. The optimal parameters for each individual element were determined by calibrating the array against an RBR Solo temperature logger. The array and the RBR Solo were both placed in a well-mixed bucket of seawater. This was done repeatedly with water temperatures ranging from about -1 °C to 10 °C in increments of about 0.5 °C. The computed temperature of each array element using the Steinhart-Hart equation was then fit to the "true" temperature measured by the RBR Solo using linear regression.



*Author contributions.* HJ, GD, and DS came up with the original idea and design for the study. DS, OG, and HJ designed and fabricated the hardware for the frame. HJ, GD, and DS designed and fabricated the instruments that were deployed on the frame. OG and HJ conducted the field campaign, making iterative improvements to the frame design and collecting the data. HJ wrote the manuscript, with assistance from GD, DS, and OG.

*Competing interests.* The authors declare that they have no conflict of interest.

*Acknowledgements.* We would like to thank Meri Korhonen for her assistance with data collection in the field, and Christian Powell for his help with fabrication of ice frame components. We would also like to thank the members of the summer 2022 Polish Polar Station expedition. This research has been supported by the National Science Centre, Poland (grant no. 2021/43/D/ST10/00616), Ministry of Science and Higher Education of Poland (subsidy for the Institute of Geophysics, Polish Academy of Sciences), and the US Office of Naval Research (grant no.
N00014-21-1-2304 and N00014-21-1-2316).



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

**Figure 1. a** A schematic of ice-ocean interactions at tidewater glaciers, including many processes that occur on scales of 10-100 m. These include subglacial discharge, water circulation in the fjord, and thermohaline structure. **b** An inset showing processes that occur within the proximal ice-water boundary, at scales ranging from roughly 0.1-100 mm. These include the release of air bubbles into the water, the rise of air bubbles through the water, and mixing within the proximal boundary. These smaller scale processes have the potential to control heat flux across the boundary and into the ice. **c** Still frames from high-speed video footage of a bubble being released from glacier ice in a laboratory experiment.




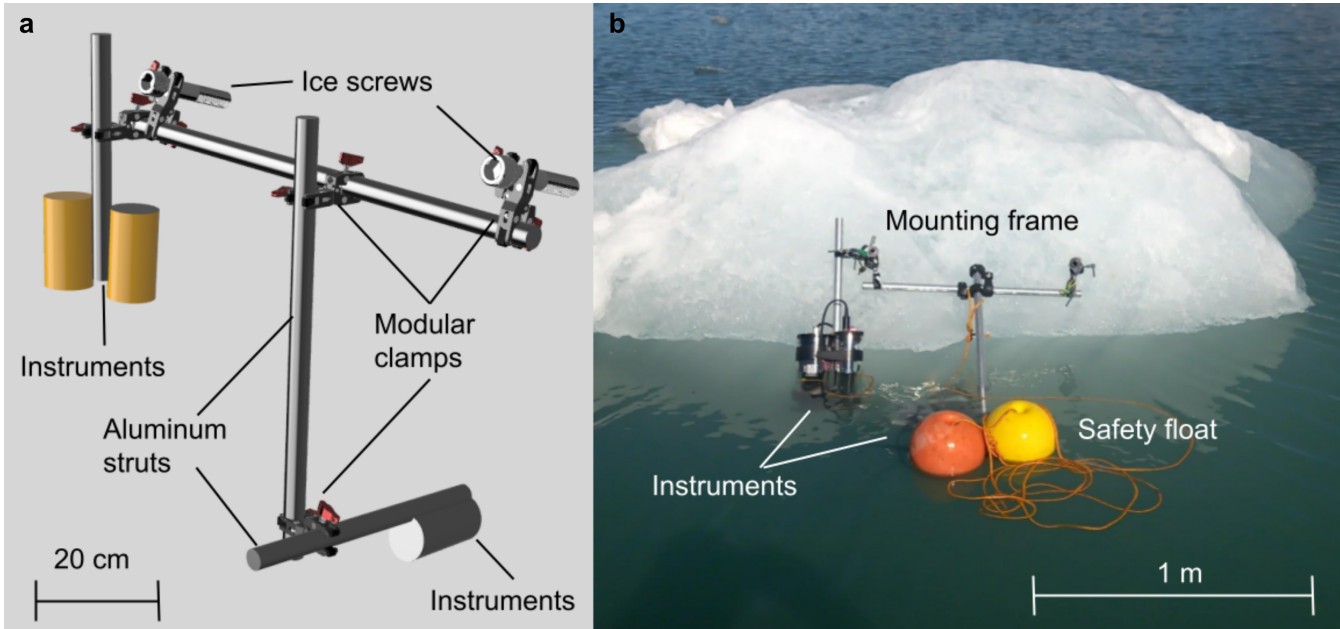

**Figure 2. a** Schematic of the ice frame. The main structure consisted of aluminum struts fastened together with a series of adjustable clamps. Two ice screws were screwed into a floating growler and the frame was attached to the screws using clamps. Instruments were fastened to the frame, allowing them to be held within the proximal boundary of floating growlers. **b** An image of the ice frame deployed on a floating growler in Hornsund Fjord, Svalbard.

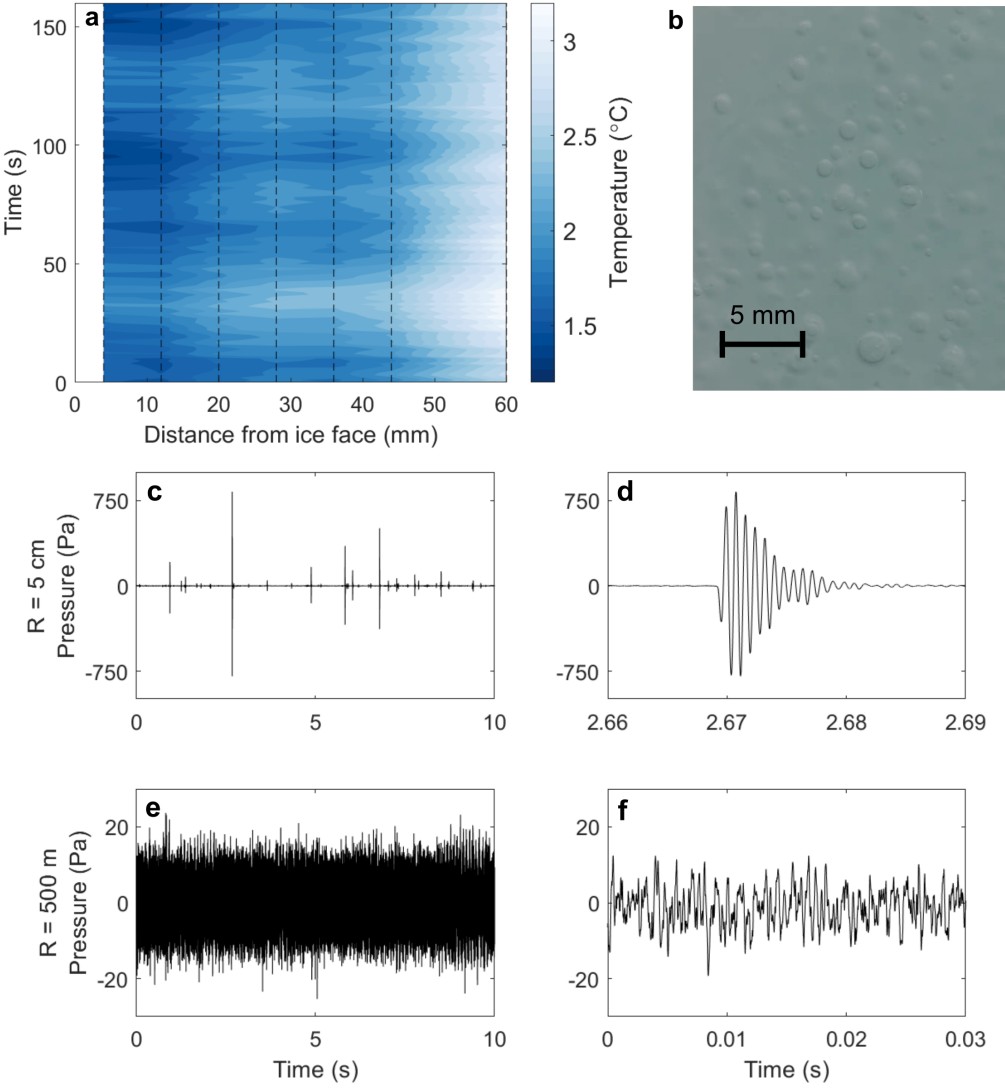

**Figure 3.** Example data collected using the ice frame. **a** Contour plot of the temperature observed in the proximal boundary of a floating growler, about 30 cm below the surface of the water. The x-axis is distance from the ice face, and the position of individual thermistor elements is shown by the vertical dotted lines. **b** Direct-view image of the ice face taken with an underwater camera, positioned about 12 cm from the ice face. Individual bubbles within the ice are visible. **c** Audio recording taken from a hydrophone mounted on the ice frame, about 5 cm from the ice face. The vertical lines are pulses generated by the release of bubbles from the ice. **d** A blown-up view of a bubble pulse recorded from the ice frame, showing a pulse from a bubble release. **e** An audio recording taken from the middle of Hansbreen bay, far from any pieces of ice, and about 500m from the glacier terminus. The majority of the signal is generated by bubbles being released from melting ice at the terminus, but many pulses are superimposed and individual pulses are indistinguishable. **f** A blown-up view of the recording from the middle of the bay, in which individual pulses are still indistinguishable.