# Peer review of "Brief communication: A technique for making in-situ measurements at the ice-water boundary of small pieces of floating glacier ice"

_The Cryosphere, 2023_

## Author Comment (AC1)

**Referee comment 1:**

Johnson et al describe an apparatus with which it is possible to obtain measurements within 0.5 m of the ice-water interface on growlers. They demonstrate the utility of this apparatus by obtaining short (less than 3 minute) time-series of water temperature and sound, as well as detailed images of the ice face, from some growlers in Hornsund Fjord, Svalbard. This apparatus permits measurements in the proximal boundary, which have previously mostly been limited to the quasi-horizontal ice-ocean interfaces or to snapshot observations further from quasi-vertical ice-ocean boundaries. The exception to this are recent observations obtained using IceFin, but such deployments are costly. As such, measurements obtained using this apparatus have the potential to improve our understanding of ice-ocean interaction in the vertical regime distal to subglacial discharge plumes, where existing parameterisations seem to perform poorly.

Whilst the apparatus has the potential to deliver important measurements, I question whether it is necessary to publish a paper dedicated to describing the apparatus and a test deployment. A lot of the manuscript is dedicated to a lengthy description of the deployment of the apparatus, which I'm sure will be useful information to anyone who uses the apparatus in future, but I'm not sure a paper is the best medium to disseminate that information. Instead, this information (and the rest of the description of the ice frame) might be better given in a users manual, which could be provided alongside a manuscript describing the important measurements and findings that the apparatus has been used to obtain (or could be used to obtain).

We appreciate the referee's comment that recent observations made under ice shelves in Antarctica using IceFin have been a great step forward in terms of making observations at ice-ocean interfaces. We will add a mention of the IceFin deployments in the description of previous observational efforts.

We understand the referee's questioning of whether it is necessary to publish a paper describing only the apparatus and methods of the ice frame. We intend to follow referee #2's suggestions to include some additional quantitative analysis of some of the data which was collected, and we would hope that this will improve the manuscript. We would also like to note that, to our understanding, reporting novel aspects of experimental methods and techniques falls within the purview of the brief communication manuscript type.

Whether or not this information remains in a paper or a user guide, I have some minor suggestions:

Line 1 and elsewhere: I think "apparatus" or similar would be more appropriate than "method". Further in the abstract the framing of the sentence around a "method" means that the ice frame doesn't really get described in the abstract and (as written) it just reads like you screwed the instruments directly to the glacier ice.

We agree that the frame itself should be referred to as something like an apparatus rather than a method. The intention with the use of the term "method" is to include also the procedures developed to deploy the frame, because we feel that describing these is important to enable others to make similar measurements successfully and safely, and without the multiple years of trial and error that we required. The point is well taken, though, and we will make an effort to distinguish more clearly in the text between the physical apparatus itself and the procedures of deploying it.

Line 27: "such processes" is a little ambiguous. Can you expand, or specify that you mean processes related to pressurized bubbles in glacier ice

Yes, we will specify that we mean processes related to the release of pressurized bubbles from glacier ice.

Line 28: "general agreement" – can you be more specific regarding the variable or behaviour here?

Yes, we will clarify that melting rates and observed boundary layer thermal and velocity structure are fairly consistent between bubble-free numerical modelling studies and bubble-free laboratory studies.

Line 29: "models also typically neglect the effects of bubbles" – consider rewording to "but neither the laboratory studies or numerical simulations include the effect of pressurized bubbles" (or similar).

Agreed.

Line 31: "submarine melting" should I think be "submarine melt rates".

Agreed.

Line 43: perhaps specify that you mean the vertical part of icebergs here, because several studies have very prolonged measurements of horizontal ice faces. Also try to be more specific than "prolonged", given that the time-series presented here are limited to a few minutes in duration.

Yes, we will specify that we mean vertical faces of icebergs, and clarify that we were able to collect up to ~1 hour of continuous data.

Section 2.1 or elsewhere: can you provide the weight of the frame somewhere?

Yes, we will specify the weight of the frame and its various components.

Section 2.1: can you provide the cost of the frame materials? (even just the total cost)

Yes, we will specify the total cost and cost of the significant components.

Section 2.1: I think the ice screw depth should be given here as well as an estimate of the melt out time during the field deployment (along with the weather conditions at the time) – could you have deployed the instruments for much longer under cooler conditions with larger ice screws?

The length of the ice screws (19 cm) is currently specified in Appendix A, but we can specify the depth that they were screwed into the ice, and also place this information in the main text. The typical time taken for the screws to melt out of 1-2 hours is currently given in section 3.1, which describes the performance of the frame. It is probably true that longer screws and cooler conditions would lengthen this time, and we can make this clear in the text.

Section 2.1: I think you should include how many attachment points for instruments are there on the current configuration of the frame? Could that be adjusted within reasonable weight constraints?

Currently there is one primary vertical beam to which instruments can be attached; the number of instruments is limited by the size and weight of the instruments. We had an additional auxiliary vertical beam, as can be seen in figure 2, which can also hold instruments. The beams are continuous, and instruments can be sampled to them at any point along their length, and there is no reason that a third beam could not be added. The tradeoff is that adding more instruments increases the complexity of the deployment, so it's difficult to specify a specific maximum number of instruments. We can add a more detailed discussion of this to the text, though.

Line 114: related to the above, "Once enough data had been gathered", is quite ambiguous. Please be more specific to your setup during this deployment.

We agree that this is quite vague, and we will clarify that the factors influencing the length of the deployment were the melting out of the ice screws, and the estimated utility of additional data.

Section 3.4: "in principle other properties of the water…" – could you also measure the distance from some point on the frame to the ice face? Could that give you a direct measure of melt rates? (similar approaches are used to measure accumulation and ablation on glacier surfaces)

This is a good point, and indeed one could measure the distance from a point on the frame to the ice face to measure ablation directly. We will add this to the possible future applications listed.

Line 207: "rods would rest against the submerged face" – I can see how this works if the iceberg tilts towards the ice frame. Does it also work if the iceberg tilts away? Can you make that clear either way in the text?

Since the fixed attachment point is at the top, gravity will pull the frame down and in towards the ice face even if it is somewhat undercut, but of course it is true that at some angle of undercut this will not work. In practice, we tended to choose growlers with relatively flat faces to mount the frame on, partly because of this very issue. We will make this clear in the text.

Figure 2 or a separate diagram: provide the measurements of the ice frame struts. Perhaps also provide the packed dimensions.

The primary frame struts are about 75 cm long, and we will specify this in the text. All of the materials for the frame fit into a Zarges box of dimensions 0.8 x 0.4 x 0.4 m, and we agree it would be wise to specify this somewhere in the manuscript as well.

Figure 3: there are some interesting fluctuations in the temperature data with a wavelength of approximately 50 seconds. Are you able to determine how they come about? Or at least demonstrate that they are not caused by movements of the growler relative to the sensors?

This is a good point. While we expect that the combination of the ice screws and gravity should be holding the sensors in a fixed position relative to the growler, we cannot be entirely certain that the observed fluctuations are not a result of the frame and sensors shifting. Alternatively, they could be a result of different water masses moving past the sensors, but since we have no observation of the flow or 3D structure of the temperature, we can only speculate. We will outline these limitations of the present data more clearly in the text.

---

## Author Comment (AC2)

**Referee comment 2:**

General comments:

Johnson et al. present a novel technique for collecting in situ measurements of the near ice-water interface of glacier ice using a frame that can be equipped with a suite of instruments and attached to the ice. Understanding heat transport through the near ice-water interface of glacier ice is key to accurately estimating submarine melting, which can be underestimated using current theory according to some observational studies. The technique of Johnson et al. seeks address a current lack of direct observations with scope to improve glacial melt estimates in the future. The authors tested their apparatus, fitted with a hydrophone, underwater camera and thermistor string, on growlers in Hornsund Fjord, Svalbard. I think this is an exciting technique and a nice, if very brief, showcase of some measurement capabilities. I think that, with some adjustments, a paper is a nice way to present this research as a proof-of-concept study.

Specific comments:

I agree with Referee #1's comment that the deployment description is quite lengthy, for example, details about boating safety and frame redeployment after unsuccessful attempts could be omitted. I also wonder if further data exploration might be feasible/possible. Temperature measurements spanning 20 minutes are mentioned in section 3.4, but only 2.5 minutes worth of data are shown. If images and audio were also collected for the 20 minutes, could you present coincident data and look for, e.g., images of bubbles being released (learning more about this seems to be a major part of the justification) corresponding to audio pulses and maybe being seen somehow in the temperature data? Some kind of quantified results from what I've mentioned above or even something like bubble release rate from the audio data that you could compare with other literature to show that your apparatus is providing novel observations would really help the discussion which is mostly rationale and outlook at the moment. The outlook is good though and proposes some interesting future applications for your system.

We thank the referee, Matthew Corkill, for the thoughtful comments.

We will go through the deployment description carefully and make an effort to trim down any superfluous information.

One important clarification, which we can and should make clearer in the manuscript, is that with the iteration of the frame described here we were able to collect acoustic and video data simultaneously, or temperature data alone. We did not manage to make all three types of measurements simultaneously.

The point about including more quantitative data analysis is nevertheless well-taken. While we do not have coincident acoustic or video data to go with the temperature data, we would be happy to incorporate some basic quantitative analysis of the temperature data, such as the average thermal gradient observed. We did not display the full 20 minutes worth of temperature

data graphically because we did not feel that doing so would provide any additional insight, as the other 18 minutes are qualitatively similar to the 2 minutes that are shown. In terms of the acoustic and video data, we can absolutely identify some individual bubbles in the video data, estimate their size and rise rate through the water, and also find the corresponding pulse generated by the release in the acoustic data, and compare its primary frequency to the natural frequency expected base on the video data size estimate.

Briefly on the frame design, could the pair of flotation spheres collide and interfere with audio data? Perhaps something like a large, sealed PVC pipe could be attached parallel to and above the crossbeam instead of the spheres (or the aluminium crossbeam could even be replaced by a larger floating one).

The concern about noise from the flotation spheres interfering with the acoustic data is valid, and in the future a solid float fixed to the frame would indeed be a good way to eliminate this issue. For the present data, the fact that the separation between the floats and the hydrophone is several meters, whereas the hydrophone is only a few centimeters from the ice face, means that spherical spreading of the sound, which results in a $1/r$ dependence for the amplitude of the signal, should reduce the impact of noise from the floats. Additionally, noise generated by the floats colliding should be at a lower frequency than the bubble pulses, making it fairly easy to differentiate and filter out if necessary.

Technical corrections:

Line 15: Consider moving details of salt and gas to the third sentence where they're mentioned and then combining the first two sentences.

Agreed, we will do this.

Line 19: Replace "this interface" with "the proximal boundary".

Agreed.

Line 35: Poorly how?

Specifically, McChonnochie and Kerr 2017 point out that the common form of the parameterizations assumes that the boundary layer is governed by a shear-driven instability, which they suggest is only valid at water velocities in excess of about ~5 cm/s, and at velocities lower than this the thickness should instead be controlled by a convective instability. They suggest that this results in an underestimate of melting in the low flow speed regime. We will make this more clear in the text.

Line 51: Please check style guide regarding capitalising figure # and appendix #.

Understood, we will correct this throughout the manuscript.

Line 107: Remove "then" after "crossbeam was".

Agreed.

Line 121: How was the size range 3-5 m determined? Also, spaces should be removed either side of dashes in this paragraph or added elsewhere.

This is the rough ranges of sizes of the growlers that we successfully deployed the frame on. As discussed around lines 84-92, growlers smaller than this generally weren't large enough to accomodate the frame, whereas larger pieces of floating ice pose more danger to the operators. Section 3.1 is intended to be a summary of the performance capabilities of the frame; if it would be useful, we can refer back to the previous text where the reasons for these limitations are explained in more detail.
We will fix the inconsistency with the dashes.

Line 133: Quantify? "Enabling calculation of a rise velocity of …".

We will add some calculated rise velocities of bubbles.

Line 141: "figure 3e-f".

We will fix this.

Line 156: "…ice face of a floating growler about 30 cm below the sea surface…".

We will make this change.

Line 183: What was the wall thickness of the tube?

The wall thickness was 0.065 inches; we will specify this in the text.

Line 185: What material was the threaded rod?

The threaded rod was stainless steel; we will specify this in the text.

Line 209: "remain"

We will fix this.

Figure 2: It would be nice to show the thermistor array and the optional standoff beam. It would also be nice to label the instruments with what they actually are.

This is a good suggestion, and we can easily add a picture of the thermistor array. The reason for the vague label "instruments" was to emphasize the fact that other types of instruments could be deployed, but we agree that it would be better to label them clearly.

Figure 3: As mentioned above, it would be nice if more data could be included here. If you have images of a bubble being released and rising, I think it might be very nice to include multiple images showing this. If the ice-face images could be linked to the audio, I would be great to timestamp the images so that they could be linked to a longer time series of audio data. I understand that this may not be possible due to distance between instruments, but that distance may be an important consideration for future deployments if the goal is to link data from all the different instruments.

As discussed in our response to the earlier comment, we will add some more quantitative analysis of the acoustic and video data. Specifically, we could present several frames of video data showing a rising bubble, as suggested, as well as the synchronous acoustic data of the bubble-release pulse.

Figure 3a: This may be easier to relate to panels c-f if time were put on the x axis (though I also understand the logic of having distance on the x axis). It might be interesting to include a black line at some specific temperature contour to better see patterns.

As mentioned earlier, the temperature data was not collected at the same time as the acoustic and video data. Ideally it would be a separate figure, but we are constrained to three figures by the brief communication format.

Figure 3d: In the caption, "A blown-up view of a bubble-release pulse recorded from the ice frame. e…"

We will fix this.

---

## Author Response (AR1)

**Referee comment 1:**

Johnson et al describe an apparatus with which it is possible to obtain measurements within 0.5 m of the ice-water interface on growlers. They demonstrate the utility of this apparatus by obtaining short (less than 3 minute) time-series of water temperature and sound, as well as detailed images of the ice face, from some growlers in Hornsund Fjord, Svalbard. This apparatus permits measurements in the proximal boundary, which have previously mostly been limited to the quasi-horizontal ice-ocean interfaces or to snapshot observations further from quasi-vertical ice-ocean boundaries. The exception to this are recent observations obtained using IceFin, but such deployments are costly. As such, measurements obtained using this apparatus have the potential to improve our understanding of ice-ocean interaction in the vertical regime distal to subglacial discharge plumes, where existing parameterisations seem to perform poorly.

Whilst the apparatus has the potential to deliver important measurements, I question whether it is necessary to publish a paper dedicated to describing the apparatus and a test deployment. A lot of the manuscript is dedicated to a lengthy description of the deployment of the apparatus, which I'm sure will be useful information to anyone who uses the apparatus in future, but I'm not sure a paper is the best medium to disseminate that information. Instead, this information (and the rest of the description of the ice frame) might be better given in a users manual, which could be provided alongside a manuscript describing the important measurements and findings that the apparatus has been used to obtain (or could be used to obtain).

We thank the anonymous referee for their thoughtful comments.

We appreciate the referee's comment that recent observations made under ice shelves in Antarctica using IceFin have been a great step forward in terms of making observations at ice-ocean interfaces. We added a mention of the icefin deployments to the introduction.

We understand the referee's questioning of whether it is necessary to publish a paper describing only the apparatus and methods of the ice frame. We have added some additional analysis of the observed temperature gradients in section 3.4 and figure 3 a-b, an estimate of the size of a bubble released from the ice in the video footage in section 3.2 and figure 3 c-d, and an estimate of the size of the same bubble based on the acoustic frequency of the pulse released in section 3.3 and figure 3 e-f.

Whether or not this information remains in a paper or a user guide, I have some minor suggestions:

Line 1 and elsewhere: I think "apparatus" or similar would be more appropriate than "method". Further in the abstract the framing of the sentence around a "method" means that the ice frame doesn't really get described in the abstract and (as written) it just reads like you screwed the instruments directly to the glacier ice.

We agree that the frame itself should be referred to as something like an apparatus rather than a method. The intention with the use of the term "method" is to include also the procedures developed to deploy the frame, because we feel that describing these is important to enable others to make similar measurements successfully and safely, and without the multiple years of trial and error that we required. We have changed the wording in the abstract and introduction to refer to the frame itself as an apparatus and draw a distinction between that and the associated methods to deploy it.

Line 27: "such processes" is a little ambiguous. Can you expand, or specify that you mean processes related to pressurized bubbles in glacier ice

Changed to "processes related to the release of pressurized gas from the ice".

Line 28: "general agreement" – can you be more specific regarding the variable or behaviour here?

Changed to "Results from numerical modelling studies on the scale of the proximal boundary have shown general agreement with some laboratory studies in terms of the observed melt rates and the structure of the thermal and velocity boundary layers"

Line 29: "models also typically neglect the effects of bubbles" – consider rewording to "but neither the laboratory studies or numerical simulations include the effect of pressurized bubbles" (or similar).

Implemented.

Line 31: "submarine melting" should I think be "submarine melt rates".

Implemented.

Line 43: perhaps specify that you mean the vertical part of icebergs here, because several studies have very prolonged measurements of horizontal ice faces. Also try to be more specific than "prolonged", given that the time-series presented here are limited to a few minutes in duration.

Changed to "Recently, robotic vehicles have been used to make observations of water properties close to the termini of tidewater glaciers, or beneath floating ice shelves, but making measurements in the proximal boundary layer of vertical ice faces remains a challenge."

Section 2.1 or elsewhere: can you provide the weight of the frame somewhere?

This information has been added to Appendix A

Section 2.1: can you provide the cost of the frame materials? (even just the total cost)

This information has been added to the appendices.

Section 2.1: I think the ice screw depth should be given here as well as an estimate of the melt out time during the field deployment (along with the weather conditions at the time) – could you have deployed the instruments for much longer under cooler conditions with larger ice screws?

The following text was added to section 3.1: "Under the conditions experienced during this field campaign, with air temperatures in the range of about 5-8 $^\circ$C and water temperatures in the range of about 2-4 $^\circ$C, and with ice screws of length 19 cm, the deployment time was limited to about 1-2 hours before melting around the ice screws would make the frame-to-growler connection unstable. This might be extended with longer screws or in colder temperatures."

Section 2.1: I think you should include how many attachment points for instruments are there on the current configuration of the frame? Could that be adjusted within reasonable weight constraints?

Currently there is one primary vertical beam to which instruments can be attached; the number of instruments is limited by the size and weight of the instruments. We had an additional auxiliary vertical beam, as can be seen in figure 2, which can also hold instruments. The beams are continuous, and instruments can be sampled to them at any point along their length, and there is no reason that a third beam could not be added. The tradeoff is that adding more instruments increases the complexity of the deployment, so it's difficult to specify a specific maximum number of instruments. The following was added to section 3.1: "The modular nature of the frame means that multiple instruments can be deployed at once, with the trade-off being increased weight and complexity."

Line 114: related to the above, "Once enough data had been gathered", is quite ambiguous. Please be more specific to your setup during this deployment.

This sentence was cut out because it wasn't adding anything to the discussion and we wanted to shorten the deployment description.

Section 3.4: "in principle other properties of the water…" – could you also measure the distance from some point on the frame to the ice face? Could that give you a direct measure of melt rates? (similar approaches are used to measure accumulation and ablation on glacier surfaces)

This is a good point, and indeed one could measure the distance from a point on the frame to the ice face to measure ablation directly. We added "In principle, other types of data could also be collected using a similar apparatus, such as water salinity or flow speed, or the ablation of the ice face itself." to section 3.1.

Line 207: "rods would rest against the submerged face" – I can see how this works if the iceberg tilts towards the ice frame. Does it also work if the iceberg tilts away? Can you make that clear either way in the text?

We added the following sentence to Appendix A: "Since the top of the vertical beam was fastened to the cross beam within 5-10 cm of the ice, as long as the ice face was roughly vertical the force of gravity would act to pull the instruments down and in towards the ice."

Figure 2 or a separate diagram: provide the measurements of the ice frame struts. Perhaps also provide the packed dimensions.

The length of the struts and packed dimensions are now provided in Appendix A.

Figure 3: there are some interesting fluctuations in the temperature data with a wavelength of approximately 50 seconds. Are you able to determine how they come about? Or at least demonstrate that they are not caused by movements of the growler relative to the sensors?

This is a good point. While we expect that the combination of the ice screws and gravity should be holding the sensors in a fixed position relative to the growler, we cannot be entirely certain that the observed fluctuations are not a result of the frame and sensors shifting. Alternatively, they could be a result of different water masses moving past the sensors, but since we have no observation of the flow or 3D structure of the temperature, we can only speculate. We added the following text to section 3.4: "There is some temporal variability that may be related to wave action or changes in water flow past the ice face, but in the absence of any additional measurements, it is difficult to determine the cause or nature of these variations. It is also possible that these fluctuations could be caused by small motions of the frame and the sensors relative to the growler."

**Referee comment 2:**

General comments:

Johnson et al. present a novel technique for collecting in situ measurements of the near ice-water interface of glacier ice using a frame that can be equipped with a suite of instruments and attached to the ice. Understanding heat transport through the near ice-water interface of glacier ice is key to accurately estimating submarine melting, which can be underestimated using current theory according to some observational studies. The technique of Johnson et al. seeks address a current lack of direct observations with scope to improve glacial melt estimates in the future. The authors tested their apparatus, fitted with a hydrophone, underwater camera and thermistor string, on growlers in Hornsund Fjord, Svalbard. I think this is an exciting technique and a nice, if very brief, showcase of some measurement capabilities. I think that,

with some adjustments, a paper is a nice way to present this research as a proof-of-concept study.

Specific comments:

I agree with Referee #1's comment that the deployment description is quite lengthy, for example, details about boating safety and frame redeployment after unsuccessful attempts could be omitted. I also wonder if further data exploration might be feasible/possible. Temperature measurements spanning 20 minutes are mentioned in section 3.4, but only 2.5 minutes worth of data are shown. If images and audio were also collected for the 20 minutes, could you present coincident data and look for, e.g., images of bubbles being released (learning more about this seems to be a major part of the justification) corresponding to audio pulses and maybe being seen somehow in the temperature data? Some kind of quantified results from what I've mentioned above or even something like bubble release rate from the audio data that you could compare with other literature to show that your apparatus is providing novel observations would really help the discussion which is mostly rationale and outlook at the moment. The outlook is good though and proposes some interesting future applications for your system.

We thank the referee, Matthew Corkill, for the thoughtful comments.

Section 2.3 has now been trimmed down substantially.

Figure 3 has been largely redone. Panel b shows the average temperature measured at each of the sensors, and the slope of the temperature gradient. Panels c and d show a bubble before and after its release from the ice, and panel e shows the acoustic pulse emitted by this bubble during its release, recorded by the hydrophone. Panel f shows the power spectrum of this pulse. The relation between the size of the bubble as observed from the video data and as inferred from the acoustic pulse is discussed in the text in sections 3.2 and 3.3.

Briefly on the frame design, could the pair of flotation spheres collide and interfere with audio data? Perhaps something like a large, sealed PVC pipe could be attached parallel to and above the crossbeam instead of the spheres (or the aluminium crossbeam could even be replaced by a larger floating one).

The concern about noise from the flotation spheres interfering with the acoustic data is valid, and in the future a solid float fixed to the frame would indeed be a good way to eliminate this issue. For the present data, the fact that the separation between the floats and the hydrophone is several meters, whereas the hydrophone is only a few centimeters from the ice face, means that spherical spreading of the sound, which results in a 1/r dependence for the amplitude of the signal, should reduce the impact of noise from the floats. Additionally, noise generated by the floats colliding should be at a lower frequency than the bubble pulses, making it fairly easy to differentiate and filter out if necessary.

Technical corrections:

Line 15: Consider moving details of salt and gas to the third sentence where they're mentioned and then combining the first two sentences.

Implemented.

Line 19: Replace "this interface" with "the proximal boundary".

Implemented.

Line 35: Poorly how?

We added the sentence: "Specifically, the existing framework implicitly assumes that the boundary layer thickness is governed by a shear instability, but at low flow speeds it may instead be controlled by a convective instability."

Line 51: Please check style guide regarding capitalising figure # and appendix #.

This has been fixed.

Line 107: Remove "then" after "crossbeam was".

Implemented.

Line 121: How was the size range 3-5 m determined? Also, spaces should be removed either side of dashes in this paragraph or added elsewhere.

The size range is explained in the third paragraph of section 2.3, and we have fixed the inconsistency with the dashes.

Line 133: Quantify? "Enabling calculation of a rise velocity of …".

We have removed this line from the text. While we could compute bubble rise velocities, it seemed like a less interesting example than the bubble size analysis currently presented, which showcases the synchronization of the audio and video data.

Line 141: "figure 3e-f".

Fixed.

Line 156: "…ice face of a floating growler about 30 cm below the sea surface…".

Fixed.

Line 183: What was the wall thickness of the tube?

The wall thickness was 0.065 inches, and this was added to Appendix A.

Line 185: What material was the threaded rod?

The threaded rod was stainless steel, and this was added to Appendix A.

Line 209: "remain"

Fixed.

Figure 2: It would be nice to show the thermistor array and the optional standoff beam. It would also be nice to label the instruments with what they actually are.

We added a panel to figure 2 showing the thermistor array, and clarified the labeling of the instruments in the figure and in the caption.

Figure 3: As mentioned above, it would be nice if more data could be included here. If you have images of a bubble being released and rising, I think it might be very nice to include multiple images showing this. If the ice-face images could be linked to the audio, I would be great to timestamp the images so that they could be linked to a longer time series of audio data. I understand that this may not be possible due to distance between instruments, but that distance may be an important consideration for future deployments if the goal is to link data from all the different instruments.

The new figure 3 panels c and d show a bubble before and after release from the ice, and panel e shows the acoustic pulse from the release of that bubble recorded by the hydrophone. Panel f shows the power spectrum of said acoustic pulse. We hope that this new figure does a better job of showcasing the data that we were able to collect.

Figure 3a: This may be easier to relate to panels c-f if time were put on the x axis (though I also understand the logic of having distance on the x axis). It might be interesting to include a black line at some specific temperature contour to better see patterns.

The temperature data is independent of the acoustic and video data, but is included in the same figure because of the 3-figure limit for brief communication manuscripts. With the addition of the average temperature plot in figure 3b, we think that the current orientation makes the most sense. Emphasizing the contour lines is a good idea, but with the dotted lines denoting the sensor positions it created a very cluttered appearance.

Figure 3d: In the caption, "A blown-up view of a bubble-release pulse recorded from the ice frame. e…"

Figure 3 and the caption are sufficiently different that this comment no longer applies.

---

## Author Response (AR2)

Dear Dr. Karlsson,

Thank you very much for your attentiveness during the review processes, and for these final suggestions. We have addressed them as follows:

1. We are citing Wengrove et al. (2023) on line 26.
2. Both instances of "unmanned" (on line 171) have been replaced with "uncrewed".

Thanks,
Hayden